



# Applying Corrective Machine Learning in the E3SM Atmosphere Model in C++ (EAMxx)

Aaron S. Donahue[1], Elynn Wu[2], W. Andre Perkins[2], Peter M. Caldwell[1], Christopher S. Bretherton[2], Finn Rebassoo[1], and Jean-Christophe Golaz[1]

[1]Lawrence Livermore National Laboratory, Livermore, CA USA
[2]Allen Institute for Artificial Intelligence, Seattle, WA, USA

**Correspondence:** Aaron S. Donahue (donahue5@llnl.gov)

**Abstract.** The Simplified Cloud-Resolving E3SM Atmosphere Model (SCREAM) is the newest addition to the family of earth system models capable of explicitly resolving convective systems. SCREAM is a kilometer-scale configuration of the advanced E3SM Atmosphere Model (EAMxx), designed for heterogeneous systems. While the enhanced accuracy of kilometer-scale modeling offers significant benefits, it comes with a substantial computational cost, limiting feasible simulation durations to only a few years, even on the fastest supercomputers. Machine learning presents an opportunity for scientists to achieve the high accuracy of storm-resolving models at a significantly reduced cost. Building on the previous success of applying corrective machine learning (ML) to the FV3 model, this study explores the effects of implementing corrective ML in EAMxx-SCREAM. We also address the computational challenges of integrating our implementation of corrective ML, which is written in Python, with the C++/Kokkos EAMxx driver, as well as potential reasons why this approach has not proved as effective for EAMxx-SCREAM as for the FV3 model.

## 1 Introduction

Accurate future climate projections are crucial to a variety of sectors, including agricultural, energy, and public health. For instance, predicting shifts in growing seasons, optimizing renewable energy sources, and understanding climate-sensitive diseases. Currently, physics-based climate models, also referred to as general circulation models (GCMs), are responsible for generating these projections and require a massive amount of computing resources in order to produce one climate realization. These models need to balance accuracy with feasibility, and most opt to use coarse spatial resolution, typically around 100 km, allowing them to produce ensembles of climate simulations for decades or centuries. However, the downside of using coarse-resolution models is their inability to resolve storms, clouds, and complex topography. The Simplified Cloud-Resolving E3SM Atmosphere Model (SCREAM) addresses the resolution problem by using kilometer-scale resolution, but it is too computationally expensive to run more than a few years at a time (Caldwell et al., 2021). This project aims to develop a computationally efficient machine learning (ML) based emulator for SCREAM, which will maintain the high accuracy of the original model while significantly reducing computational costs. By leveraging ML techniques, the emulator can be trained to replicate the behavior of SCREAM at a fraction of the computational expense.



ML has the potential of revolutionizing how weather forecasts and climate predictions are generated. Within this active area
of research, there are two primary approaches. The first approach relies solely on ML models trained on historical observations,
reanalysis datasets, or outputs from existing GCMs (Keisler, 2022; Chen et al., 2023; Lam et al., 2023; Price et al., 2023). The
second approach involves a hybrid framework that combines machine learning with traditional GCMs. In this method, machine learning methods are often used to replace a specific physical process of GCMs (Kochkov et al., 2023; Henn et al.,
2024) or to apply column-wise correction to the coarse-grid model (Bretherton et al., 2022). Both approaches have shown
successful results, often outperforming state-of-the-art GCMs while only using a fraction of the computational resource. This
study builds on the success of a similar study by the Allen Institute for Artificial Intelligence (Ai2) working with the FV3GFS
model (https://github.com/ai2cm/fv3gfs-fortran). Ai2 was able to improve the predictive accuracy of relatively coarse resolution global climate simulations using a machine-learning based correction to the model state (Bretherton et al., 2022; Kwa
et al., 2023; Sanford et al., 2023).

This study applies their corrective ML approach to SCREAM. Section 2 provides the recipe for the development of the ML
training data. Section 3 covers how the corrective ML model was embedded into the SCREAM code base. Section 4 examines
how well the ML corrected coarse model performs with respects to the fine resolution target, followed by discussion of the
results and conclusions of the study in sections 5 and 6.

## 2 Methods

SCREAM is a configuration of the atmosphere component of the Energy Exascale Earth System Model (E3SM) targeting
kilometer scale global resolutions. In order to accomplish performant simulations at this scale, the E3SM Atmosphere Model
(EAMxx) was rewritten from the original Fortran code to C++/Kokkos (Donahue et al., 2024). The adoption of Kokkos in
EAMxx unlocks the computational power of mixed CPU/GPU machines and ensures that EAMxx maintains good performance
across a number of high performance computing systems.

While EAMxx achieves unprecedented computational performance for a km-scale model (Taylor et al., 2023), its computational cost is still prohibitive for global km-scale simulations of a decade or more in length. This limitation makes it attractive
to apply corrective ML to EAMxx in order to represent effects of fine scale features at coarse resolution. We follow Ai2's approach with FV3GFS, using a km-scale EAMxx simulation to produce highly accurate training data for a corrective ML model
embedded in coarse resolution EAMxx simulations. The approach is modified to handle technical and architectural differences
between FV3GFS and EAMxx.

The corrective ML approach described in this study can be represented by four steps (Figure 1). The first step involves
the generation of a fine-grid reference simulation. This simulation serves two purposes: to provide the nudging targets for
the training data set and to provide a validation data set for the ML-corrected solution. The second step involves nudging
an EAMxx simulation at the target coarse resolution to the reference state. 'Nudging tendencies' derived from this step are
used for the third step, training of the corrective ML models. Finally, a coarse-grid model is run with added inline corrections



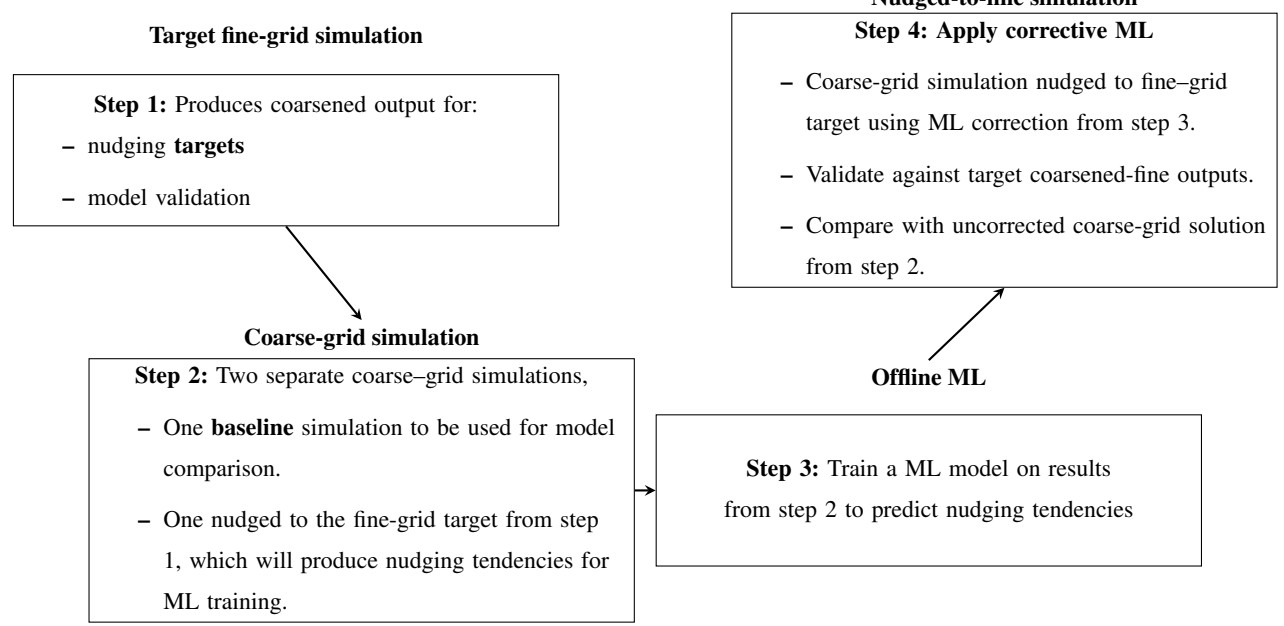

**Figure 1.** Flow diagram for the corrective ML method.

from these ML models; ideally the resulting ML-corrected coarse model will more closely approximate the time evolution and climate of the target fine-grid model. Each of these steps is described in more detail in the following subsections.

## 2.1 A km-scale reference solution

For the training and validation discussed in this paper, we used one year of customized outputs from a reference fine-grid
simulation using the default SCREAM configuration of EAMxx (Donahue et al., 2024). This used a cubed–sphere spectral element grid for the dynamics calculation which had a **n**umber of **e**lements per cube face of 1024×1024. Each element contains a **p**hysics **g**rid with a 2×2 arrangement of columns. Referred to in shorthand as an 'ne1024pg2' horizontal grid, this grid has approximately 3.25 km horizontal resolution, 128 vertical levels, and a physics timestep of 100 seconds. Notably, the SCREAM configuration omits a deep convection parameterization, regardless of grid resolution.

This was the 'standard climate' component of a Cess-style SCREAM simulation pair further discussed by Terai et al. (2025). It started on August 1, 2019 and used a climatological seasonal cycle of sea-surface temperature (SST) and sea ice.

Temporally averaged model output was produced every 3 hours. Three-dimensional fields required for our ML study were vertically interpolated natively in SCREAM onto a set of fixed pressure levels (specified below), then horizontally averaged (area-weighted) along pressure surfaces to a target 'ne30pg2' (≈ 100 km) coarse grid, masking fine-grid points below the sur-
face. Two dimensional ML-relevant fields such as surface precipitation were also horizontally averaged inline to the 'ne30pg2' grid. Only these horizontally coarsened 3D and 2D outputs were stored. See Tables 1 and 2 for a list of output variables.





The vertical grid used for pressure interpolation has 221 vertical levels. Starting with a lowest level of 1075 hPa, each subsequent level decreases by 5 hPa until 540 hPa. From 540 hPa to 270 hPa the spacing is 10 hPa, and from 270-145 hPa the spacing is again 5 hPa. For the remainder of the upper atmosphere the vertical grid uses the upper 60 typical isobaric vertical

layers in an EAMxx grid, for a total of 221 layers. The interpolated data is masked for below-surface pressure levels, which is particularly important in mountainous terrain. The horizontal area-weighted averaging from the fine to the coarse grid is performed using the Tempest remap algorithm (Ullrich and Taylor, 2015; Ullrich et al., 2016), which efficiently redistributes data across different grid resolutions.

The approach to the vertical remapping in this study is somewhat different than in the FV3GFS study. There, fine-grid

data were vertically interpolated to the local pressure levels of the coarse resolution data, masking out fine-grid data at any resulting grid points below the surface (since the FV3GFS model levels are terrain-following, these pressure levels differ between coarse grid cells). They then used the same method for horizontal coarsening as we do. Our coarse-grid ne30pg2 EAMxx configuration has only 72 terrain-following vertical levels, rather than the 128 levels of the fine-grid configuration. In the FV3GFS application, the coarse and fine-grid model versions had the same set of vertical levels.

To maintain hydrostatic balance and conserve atmospheric mass during the vertical interpolation and horizontal coarsening process, the FV3GFS study also applied a small correction to the area-weighted surface elevation, which we neglect. See section 2.4 in Bretherton et al. (2022) for more details.

For the FV3GFS study, instantaneous 3-hourly snapshots of the prognostic fields were stored, rather than the 3-hour average fields here. Our approach produces smoother nudging tendencies; the FV3GFS approach has the interpretational advantage

that 3-hourly average tendencies are just differences between successive 3-hourly snapshots.

**Table 1.** List of variables used for nudged runs

| Variable Common Name | EAMxx Variable Name | Units | Dimension | |
|---|---|---|---|---|
| Temperature | T_mid | K | 3D | Nudged |
| Specific Humidity | qv | kg/kg | 3D | Nudged |
| Eastward Wind | U | m/s | 3D | Nudged |
| Northward Wind | V | m/s | 3D | Nudged |
| Surface near-IR direct flux | sfc_flux_dir_nir | W/m2 | 2D | Prescribed |
| Surface UV/visible direct flux | sfc_flux_dir_vis | W/m2 | 2D | Prescribed |
| Surface UV/visible diffuse flux | sfc_flux_dif_vis | W/m2 | 2D | Prescribed |
| Net (down - up) SW flux at surface | sfc_flux_sw_net | W/m2 | 2D | Prescribed |
| Downwelling LW flux at surface | sfc_flux_lw_dn | W/m2 | 2D | Prescribed |
| Liquid precipitation flux | precip_liq_surf_mass_flux | kg/m2 | 2D | Prescribed |





**Table 2.** List of variables used for validation of ML model

| Variable Common Name | EAMxx Variable Name | Units | Dimension |
|---|---|---|---|
| Water vapor path | VapWaterPath | mm | 2D |
| Total precipitation to the surface | precip_liq_surf_mass_flux + precip_ice_surf_mass_flux | mm/day | 2D |
| Surface temperature | surf_radiative_T | K | 2D |
| Top of atmosphere upwelling LW | LW_flux_up_at_model_top | W/m2 | 2D |
| Top of atmosphere upwelling SW | SW_flux_up_at_model_top | W/m2 | 2D |
| Sensible heat flux | surf_sens_flux | W/m2 | 2D |
| Latent heat flux | surf_evap | W/m2 | 2D |
| Surface SW net flux | sfc_flux_sw_net | W/m2 | 2D |
| Surface downwelling LW flux | sfc_flux_lw_dn | W/m2 | 2D |

## 2.2 Machine Learning Training Data - Nudged Simulation

Similar to the prior FV3GFS work, our training dataset for corrective ML was constructed from a coarse-resolution ne30pg2 simulation that was nudged to track the vertically interpolated and horizontally coarsened reference solution evolution described in section 2.1. At the end of each atmospheric timestep, the model state was adjusted to match the fine-resolution solution state using data from the reference simulations. As in prior work, the nudging timescale was set to 3 hours. At each timestep, the reference data was interpolated to the current model time before nudging.

EAMxx employs a hybrid pressure coordinate system vertically (Dennis et al., 2005), thus the reference data also required vertical interpolation from its fixed vertical pressure grid onto the model's pressure coordinates at each timestep. For the training dataset simulation, only the three-dimensional variables—temperature, specific humidity, and horizontal velocities—were nudged. The other surface variables, as detailed in Table 1, were directly prescribed at the end of the atmospheric timestep. The nudging tendencies from this simulation were averaged every 3 hours and provided as output. In addition, the training data requires a record of the model state that led to each nudging tendency, recorded at the same 3-hourly times as the tendencies.





## 2.3 Training the Model

Following Ai2's prior work with FV3GFS, the nudging tendencies recorded during the nudged simulation discussed in section
2.2 are used to train three separate corrective ML models for different state variables in EAMxx. All models utilize Ai2 Climate
Modeling Group's open-source `fv3net` Github repo. The training process is column–independent, meaning that each training
data point does not consider neighboring spatial columns in the grid. However, the training dataset does include some grid
location information through the latitude and cosine zenith angle associated with each point. Below, we outline the parameters
for each model.

1. **Thermodynamic Model:** This model focuses on the thermodynamic state of the atmosphere. The input variables for
training include temperature, specific humidity, latitude, surface geopotential, and cosine zenith angle. The training
outputs are the prescribed nudging tendencies for temperature and specific humidity.

   2. **Momentum Model:** This model targets the atmospheric momentum state. The outputs are the eastward and northward
velocity nudging tendencies. The input variables are the same as those used in the thermodynamic model, with the
addition of eastward and northward velocity.

   3. **Surface Forcing Model:** This model addresses the surface forcing passed to other components of the earth system.
Unlike the other models, it does not use nudging tendencies as input. Instead, it relies on the prescribed surface fluxes
calculated in the reference solution. The inputs for this model match those of the thermodynamic model, while the outputs
are the surface downwelling longwave flux and surface net shortwave flux broken down into near-infrared and visible
direct and diffuse flux fraction. We apply output limiters to ensure physical realizability (e.g., downward near-infrared
diffuse fraction is between 0-1).

All models are trained for 300 epochs, with an early stopping condition if convergence is detected. Random subsets of dates
from the EAMxx dataset are selected for training and validation.

## 2.4 Coarse-grid Model

The three machine learning models developed in section 2.3 are integrated into EAMxx to create a coarse-grid model that
aspirationally produces solutions closer to fine-grid accuracy. For more details on how the ML models were embedded in the
EAMxx code base please refer to the section on Python/C++ coupling, section 3. The corrective ML is applied at the end of
each model timestep, with the thermodynamic and momentum models operating independently but using the same model input
state. The surface forcing model updates the surface fluxes based on the ML-corrected state.
Special care is needed when applying corrections from the thermodynamic model. Nudging specific humidity effectively
adds or removes water vapor mass from the system, which typically reflects precipitation differences in the fine-resolution
solution compared with the coarse–resolution solution. To account for this, the total change to the column water mass is
calculated after adjusting specific humidity, and this result is then added to or subtracted from the total liquid precipitation. A





mass clipper is employed to ensure that precipitation values remain non-negative. This correction method does not account for
ice precipitation. Future studies could explore separate ice and liquid precipitation adjustments, depending e.g. on near-surface
air temperature.

Likewise, the adjustment to specific humidity does not correct the cloud state toward the reference solution. Coarse-model
cloud biases affect the accuracy of coarse-model radiative forcing to the surface models, which could feed back on the atmo-
spheric evolution. Since our corrective ML model directly learns the surface fluxes from the fine-grid reference data rather than
predicting corrective fluxes, we expect it to have surface fluxes consistent with the fine-grid reference simulations given the
input profiles.

## 3    Implemention of ML workflow in EAMxx (Python/C++ coupling)

We opted to utilize an existing framework, `fv3net`, built by Ai2 and used in Bretherton et al. (2022); Clark et al. (2022); Kwa
et al. (2023). `fv3net` includes all aspects of the ML workflow, including pre-processing native model output, ML training,
offline and online testing, and reporting. `fv3net` was originally developed for Geophysical Fluid Dynamics Laboratory's
(GFDL) FV3GFS model. We made necessary modifications to work with EAMxx's data, and all updates are available in the
open-source `fv3net` repository.

One of the main challenges is that `fv3net` is written in Python whereas EAMxx is in C++. We use `pybind11` to bridge
between C++ and Python (W. Jakob, J. Rhinelander, D. Moldovan and others, 2017). The strategies for running on CPU and
GPU are slightly different. On CPU, we utilize `pybind11`'s numpy bindings to transfer EAMxx's Kokkos view data to Python
and declare them as numpy arrays. We do this as a non-copy operation, making the operations relatively cheap. On GPU, the
data passing is different since numpy does not support GPU. In this case we pass EAMxx's Kokkos view data as pointers to
Python and reconstruct numpy-like arrays from the Cupy package through unmanaged memory access (Okuta et al., 2017). As
in the CPU case, we do not do any memory copies. Once Cupy arrays are constructed on the Python side, we are able to reuse
the same workflow as in the CPU case. For both the CPU and GPU implementation the state variables are directly overwritten
during the Python calls. The ML workflow adds 10% of overhead for the CPU run and 6% for GPU. There is also a one-time
initialization cost for loading the necessary libraries in Python, which becomes negligible once the simulation is sufficiently
long (e.g., 1 simulation month for the 100km resolution case).

While it is possible avoid Python and instead directly interface with a trained ML model in C++ (e.g. via Tensorflow's C++
interface), our approach allowed us to take advantage of the existing `fv3net` package's robustness and stability. For example,
we are able to take advantage of safeguards built into `fv3net` to ensure ML has physically-consistent output. Since `fv3net` is
an end-to-end ML workflow, we also utilize its reporting for data analysis and visualization. Overall, using `fv3net` drastically
reduced development time in every step of the ML workflow.



## 4    Results from km-scale experiment

Results from a one-year free-running coarse-grid simulation including 'online' ML correction toward the fine-grid reference are shown (Donahue and Wu, 2025). We use an uncorrected ne30pg2 run as a baseline comparison. ML correction tends to underpredict nudging tendencies for both temperature and specific humidity. Figures 2 and 3 show the annual-mean bias and root mean square error (RMSE) of baseline and ML-corrected runs from the reference data (3.25 km coarsened to 100 km, also used as ML training data). For this experiment, we found the largest improvement in surface downwelling longwave flux, with

16% improvement in global RMSE when compared to the baseline. Additionally, we found marginal improvement (5-10%) for net shortwave flux at the surface, precipitation, and surface pressure. However, RMSE was worse for surface temperature (24%) and total water path (43%) in the ML-corrected run.

To evaluate the stability and robustness of the corrective ML approach, we conducted a series of trainings using different random seeds in the neural networks. During the online prognostic runs, some seeds crashed due to the inputs being out-of-

sample. To address this issue, we adopted the method outlined in Sanford et al. (2023), training a separate out-of-sample novelty detector to complement the existing neural networks for temperature and specific humidity modeling. This approach proved reliable, allowing all random seeds to successfully complete a full-year simulation. A sample of these trainings is presented in Figure 4. As shown, the solution exhibits some dependence on the random seed and the use of novelty detection. Clear regional differences are evident between models trained with and without novelty detection. For instance, models trained with novelty

detection produced higher overall longwave downwelling fluxes.

## 5    Discussion

While we adopted the same strategy as Ai2, applying the workflow to a different atmospheric model, it quickly became clear that implementation would be challenging due to the different details in each model. Each step in the corrective ML workflow— nudging from coarse to fine resolution simulation, prescribing surface fluxes for the coarse simulation, integrating ML, and

diagnostics— is tightly coupled to the specifics of the underlying model, making generalization difficult. Even though this project was a collaboration between the team that developed EAMxx and the team that successfully applied this corrective ML approach to fv3gfs, the end result fell short of the desired improvement from corrective ML.

There are several possibilities for why corrective ML was less successful for EAMxx than it was for FY3GFS. The most obvious difference between the 2 modeling frameworks is that the ML-corrected version of EAMxx doesn't include a deep

convective parameterization and hence looks less realistic. As a result, more is being asked of corrective ML in EAMxx than in FV3GFS. Our initial expectation was that larger differences between the nudged and free-running low-resolution simulation would provide more signal for training the ML. But in such an environment, imperfections in the corrective ML will also stand out more. It may be that improvements in ML due to stronger signal are too small to overcome the increased need for corrections in EAMxx. Future work will address this hypothesis when EAMxx is enhanced to support a low-resolution

configuration with parameterized deep convection.



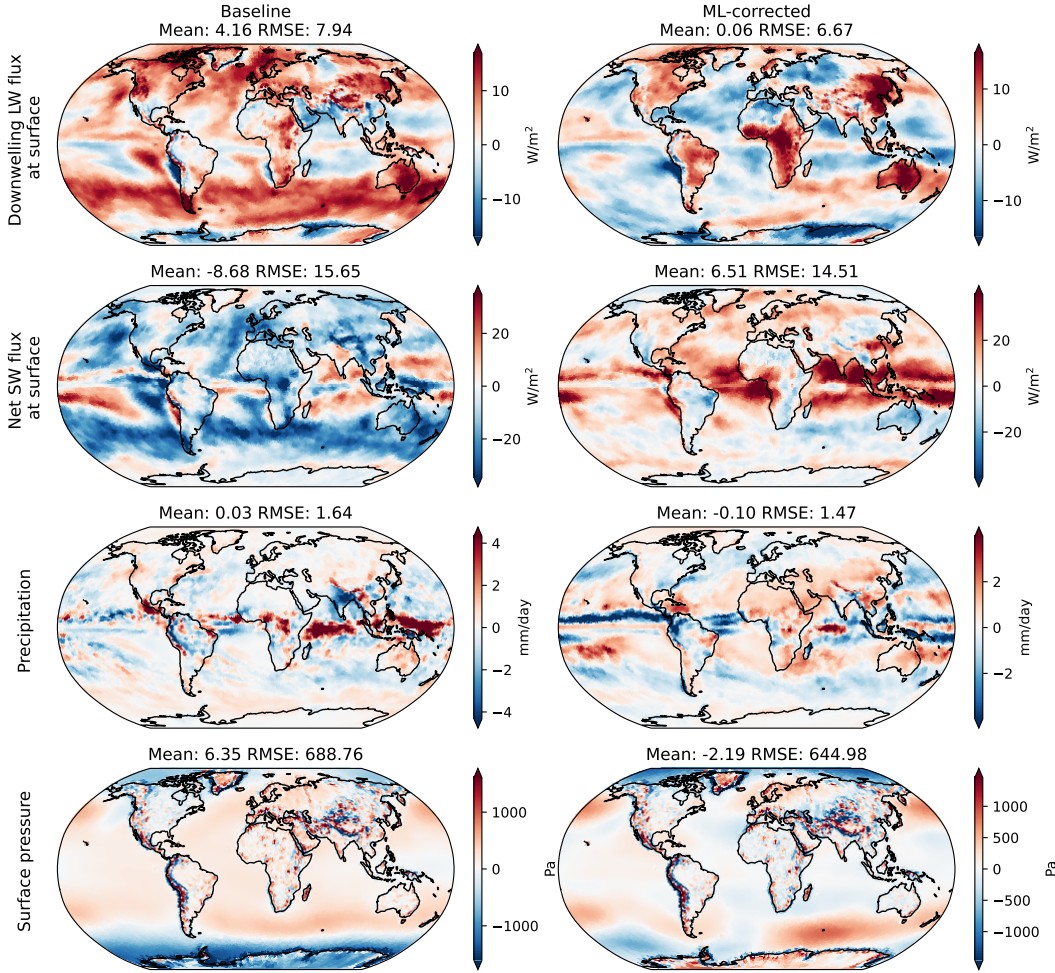

**Figure 2.** Spatial pattern of annual mean simulation error with respect to the coarsened fine-grid reference simulation. As compared with the uncorrected coarse-resolution baseline (left column), the coarse simulation including corrective ML (right column) reduces the RMSE for several surface variables, such as precipitation and surface pressure.

As discussed in the Methods section, there are three independent ML models that make up the full corrective ML implementation, each with its own set of parameters to train on. In this study we focused on training the models following the same approach as with the `fv3gfs` test case. Given the differences between EAMxx and `fv3gfs` it is possible that a successfully trained model may require an adjustment to the training hyperparameters. An anecdotal example is the use of latitude as an input into the ML training data set. We found that the inclusion of latitude in the training input improved the accuracy of the model, presumably as a proxy for physical variables that correlate strongly with latitude but are not well represented in the coarse model. A subsequent study could investigate what other parameters and input variables improve the performance of the models.





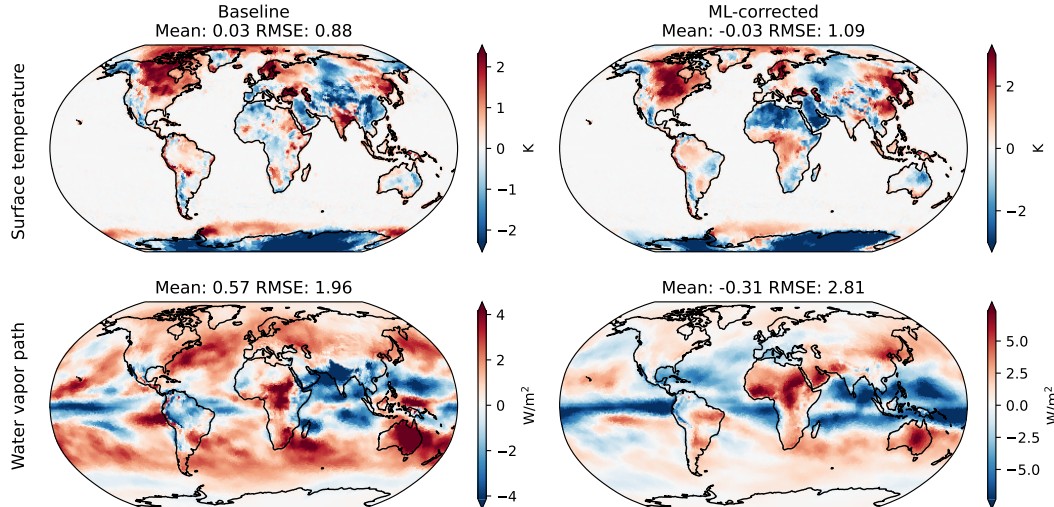

**Figure 3.** As compared with the uncorrected coarse-resolution baseline (left column), the coarse corrective ML simulation (right column) increases the RMSE of annual-mean spatial patterns for surface temperature and total integrated water vapor.

Adoption of the corrective ML framework involved the development of several new features in EAMxx. EAMxx is heavily
unit tested, but there is always a chance that a difficult bug or untested edge case is able to make it past the testing. There are three major developments that were added to EAMxx as part of this effort; Python to C++ bridging for ML, ability to nudge the model state, and prescribed atmospheric surface fluxes. While significant effort went into developing and testing these new features, future work may want to revisit their implementation as a potential source of error in the ML implementation.

Finally, as outlined in the Methods section, the fine-resolution solution was coarsened to the target resolution both hori-
zontally and vertically. The decision to map onto fixed vertical pressure coordinates resulted in situations where grid points over topography could lead to many or all data points being masked. This could create either masked data in the target state or grid points with disproportionate weighting based on only a few source columns. To investigate this hypothesis, a separate lightweight study was conducted using a 25 km fine-resolution simulation, comparing two corrective ML models. One model utilized the approach described in this study, while the other did not apply vertical interpolation for the target state. The latter
model showed slightly better performance. Unfortunately, due to the high cost of 3.25 km simulations, it was not feasible within the scope of this project to repeat this study using horizontal-only interpolation.

## 6    Conclusions

Although this work did not achieve the anticipated accuracy of the corrective ML version of EAMxx, it did have several beneficial effects. In particular, this study marked the first time Python packages were integrated into EAMxx, providing a
blueprint for future hybrid ML without degrading EAMxx performance on either CPU or GPU architectures. This work also





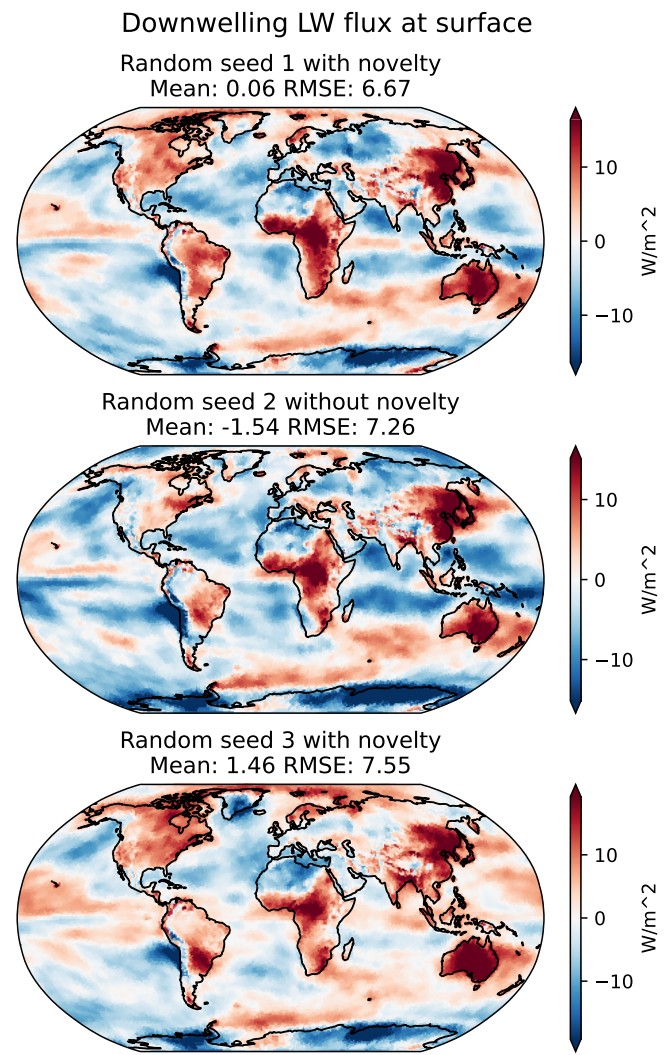

**Figure 4.** Yearly averaged downwelling longwave flux at the surface bias between ML corrected runs and coarsened-fine target (3.25km to 100km). Three random seeds shown, seed 1 and 3 use an additional novelty detection model, while seed 2 does not.

led to the development of new infrastructure within EAMxx to support prescribed surface fluxes and model nudging. While the latter was already a planned milestone for the regionally-refined configuration of EAMxx, this effort expanded that capability. The introduction of prescribed surface fluxes is important for some doubly-periodic use cases and is useful for debugging and hypothesis testing. Finally, the fact that EAMxx obtained less benefit from corrective ML than FV3GFS is a useful data point 225 in the quest to understand generalizability of ML approaches in climate modeling.

There is increasing interest in the earth system modeling community in utilizing differentiable algorithms and software. This strategy has already been successfully applied to NeuralGCM, which marries a spectral dycore coded in the Jax machine

learning language with a learned ML representation of the combined physical parameterizations of a global model, trained to evolve following a gridded reanalysis (Kochkov et al., 2024). In this setting, the ML model can be trained end-to-end including feedbacks with other model components within this differentiable framework, dramatically improving the prognostic performance of this hybrid model such that it has highly accurate weather forecast and climate skill. In principle, an analogous approach could be implemented to dramatically improve the prognostic skill of corrective ML within EAMxx. However, refactoring EAMxx to enable such a capability while retaining leadership-class performance would be a substantial software engineering challenge.

*Code and data availability.* The EAMxx model code used in this study is open–source and available at https://github.com/E3SM-Project/E3SM. The fv3net model used to train the corrective–ML model is open–source and available at https://github.com/ai2cm/fv3net. The high resolution data used in this study came from (Terai et al., 2025), instructions for accessing this data is described in this manuscript. All data used for analysis and figure generation is publicly available at Donahue and Wu (2025), doi: https://doi.org/10.5281/zenodo.16803929

*Author contributions.* AD, EW and WP wrote and tested the model code. AD and EW prepared the training and validation data sets. CB, CG, PC and FR supervised experimental design and contributed to the interpretation of the model results. AD and EW wrote the paper with feedback from WP, CB, CG, PC and FR.

*Competing interests.* Some authors are members of the editorial board of journal GMD.

*Acknowledgements.* This work was performed under the auspices of the U.S. Department of Energy by Lawrence Livermore National Laboratory under Contract DE-AC52-07NA27344. IM Release number LLNPUB-2006061. This research used resources of the National Energy Research Scientific Computing Center (NERSC), a Department of Energy Office of Science User Facility using NERSC award BER-ERCAPm4492. The authors acknowledge the use of an AI language model to assist with grammar correction and sentence structure improvements during manuscript preparation.



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
