# Peer review of "Applying Corrective Machine Learning in the E3SM Atmosphere Model in C++ (EAMxx)"

_EGUsphere, 2025_

## Referee Comment (RC1)

**Review of "Applying Corrective Machine Learning in the E3SM Atmosphere Model in C++ (EAMxx)" by Donahue et al.**

**General review:** The paper is in general well written and well structured. It provides interesting findings about transferring a concept of a machine learning based correction scheme of subgrid source terms from the FV3GFS general circulation model to the E3SM general circulation model The content of this paper is helpful for the modeling community as it clearly states encountered. problems of such a migration. Despite this I recommend the authors to expand the interpretation of the results of their hybrid simulations. Moreover all figures require an improved description what is shown in the respective captions. Likewise I strongly recommend to the authors to adjust their way how they cite data or software used to fulfill the standards of Copernicus journals.

As a result I recommend major revision of this submitted manuscript. Below you will find some specific comments and a detailed list of technical comments with respective line numbers in the submitted manuscript.

**Specific comments:**

Figures and tables: I strongly recommend to revise the captions of all Figures. There are some clear inconsistencies how you name the error metrics in the captions of the figures. e.g. Fig. 2, Fig.3 and Fig.4.  Also the shown subfigures and what its is shown needs to be explained more in detail. In addition, I would suggest to remove sentences that are evaluating the results from figure captions.

GCMs vs. neural networks:  The use of „model" for both a GCMs and neural networks complicates the understanding of the manuscript for a broader audiences. In this case you need to find a consistent and clear nomenclature throughout the manuscript. Otherwise it is very hard to understand whether you mean a neural network or a GCM for a reader.

Interpretation of the results of the conducted simulation:
Your drawn conclusion from your very short results section are important for the community as they provide helpful information for follow-up experiments from other working groups. Nevertheless your results section needs more description of the results. The shown Figure 2 and 3 provide more details that need to be explained and compared, e.g. the bias in the area of the intertropical convergence zone that led to your conclusions or where the ML-corrected simulation is clearly beneficial, e.g. the shortwave and long wave radiation biases over the Southern Ocean. Also I would suggest to add a figure that shows zonal means in a latitude-height plot of e.g. temperature differences and humidity differences. This will help the reader to understand the drawn conclusion in your discussion and conclusion section.s

Citations: Below you will find some suggestion for citations that you may add. I feel it very important to pay attention on the way you are referring to your own work. You are saying, e.g. based on the „AI2's approach", but there you need to add a citation for people that are unfamiliar with previous work. The same applies when you are referring to „FV3GFS" studies or the neural networks you are applying. It would be good to give references in this case for readers that are unfamiliar with existing work.

**Technical comments:**

Here are some technical comments. No need to address them fully, but please take them into consideration to facilitate the understanding of your interesting submitted manuscript.

Line 3: „designed for heterogenous systems" feels unclear. Here I would like to suggest to add a little bit of guidance for readers that come from a different scientific background, something along the line „heterogenous high performance computing systems". Otherwise there may be some misunderstanding whether you still mean the „Earth system" here from the previous sentence.

Line 5: Instead of a „few years" I would prefer to say „decades" as some European research projects recently run experiments with km-scale models over a few tens of years successfully, e.g. NextGems (see the Hohenegger 2023 below) or EERIE.

Line 10: Please harmonize how you refer to „FV3" or „FV3GFS" (see line 31) throughout the manuscript.

Line 15: Here you may add some references as others draw a similar conclusion in the past, e.g. Hohenegger2023 as a recent example.

Hohenegger, C., Korn, P., Linardakis, L., Redler, R., Schnur, R., Adamidis, P., Bao, J., Bastin, S., Behravesh, M., Bergemann, M., Biercamp, J., Bockelmann, H., Brokopf, R., Brüggemann, N., Casaroli, L., Chegini, F., Datseris, G., Esch, M., George, G., Giorgetta, M., Gutjahr, O., Haak, H., Hanke, M., Ilyina, T., Jahns, T., Jungclaus, J., Kern, M., Klocke, D., Kluft, L., Kölling, T., Kornblueh, L., Kosukhin, S., Kroll, C., Lee, J., Mauritsen, T., Mehlmann, C., Mieslinger, T., Naumann, A. K., Paccini, L., Peinado, A., Praturi, D. S., Putrasahan, D., Rast, S., Riddick, T., Roeber, N., Schmidt, H., Schulzweida, U., Schütte, F., Segura, H., Shevchenko, R., Singh, V., Specht, M., Stephan, C. C., von Storch, J.-S., Vogel, R., Wengel, C., Winkler, M., Ziemen, F., Marotzke, J., and Stevens, B.: ICON-Sapphire: simulating the components of the Earth system and their interactions at kilometer and subkilometer scales, Geosci. Model Dev., 16, 779–811, https://doi.org/10.5194/gmd-16-779-2023, 2023.

Line 18: Here you may need to add a reference as well, e.g. the Satoh2019 SRM paper

Satoh, M., Stevens, B., Judt, F., Khairoutdinov, M., Lin, S., Putman, W. M., and Duben, P.: Global cloud-resolving models, Current Climate Change Reports, 5, 172–184, 2019.

Line 24: Here you may add a reference to Eyring2024, as they draw a similar conclusion

Eyring, V., Collins, W. D., Gentine, P., Barnes, E. A., Barreiro, M., Beucler, T., ... & Zanna, L. (2024). Pushing the frontiers in climate modelling and analysis with machine learning. *Nature Climate Change*, *14*(9), 916-928.

Line 29: Here you may add a reference to Hu2025, as this is an example that replaces an MMF setup in E3SM.

Hu, Z., Subramaniam, A., Kuang, Z., Lin, J., Yu, S., Hannah, W. M., et al. (2025). Stable machine-learning parameterization of subgrid processes in a comprehensive atmospheric model learned from embedded convection-permitting simulations. Journal of Advances in Modeling Earth Systems, 17, e2024MS004618. https://doi.org/10.1029/2024MS004618

Line 31: Please introduce the acronym FV3GFS and provide also a reference to a paper here e.g.

Zhou, L., Lin, S.-J., Chen, J.-H., Harris, L. M., Chen, X., and Rees, S. L.: Toward Convective-Scale Prediction within the Next Generation Global Prediction System, B. Am. Meteorol. Soc., 100, 1225–1243, https://doi.org/10.1175/BAMS-D-17-0246.1, 2019.  a

Line 45: I would suggest to rewrite this sentence as it may feel potentially slightly subjective. I agree that a throughput of more than 1SYPD is quite a gain in computation efficiency but this does not suffices a detailed model simulation evaluation to claim something as „unprecedented". I would suggest just saying that : „EAMxx achieved to simulate more than a 1 year per day in km-scale simulations (Taylor 2023),…"

Line 48: I would suggest to add a reference after „AI2's approach (e.g. Bretherton2022)" to give one example reference for people that are new to the topic of hybrid modeling. Also I would suggest to replace „highly accurate training data" with for example just „fine-scale training data" to help the readers get your main point (That you are learning from high resolution data which includes fine-scale structures that a GCM would not be able to simulate directly).

Line 49: This sentence may feel a little bit unclear for readers that are not familiar with the topic. Please give here one example what you modified or stress that you will explain the modifications in the following subsection 2.1 or 2.2.

Line 55: I would suggest to write here a „coarse EAMxx simulation is run with inline.." instead of „coarse-grid model" to increase the clarity of the sentence.

Figure 1: This figure needs some major revisions. You need to highlight at least the different GCMs (e.g. coarse EAMxx, SCREAM) you are using for each step in such a central introduction figure. Otherwise it is very hard to understand for readers that are only scanning over the manuscript what „Coarse-grid simulation" is.  Also try to use a similar offset between the headers of each step and the respective frame. The same applies to the length of the different arrows. Furthermore I would suggest to give more explaining details in the caption of the figure to make it easier to understand for readers. Something a long the line „Flow diagram for the corrective ML approach, which uses ML predicted nudging tendencies (step 3) to correct coarse scale EAMxx simulations (step 4).

Line 64: Please add a citation here, where interested readers can find this essential information about omitting a deep convection parameterizations regardless of resolutions!

Line 71: Please give some details what Table 1 and Table 2 contains in a dedicated sentence for each table.

Line 79: I would delete the „somewhat" between „is" and „different" as it does not add additional information for the reader. Please add a reference at the end of the sentence to make clear against which of the few FV3GFS studies you will draw a comparison in the following sentences.

Tables 1, 2: Please add some descriptive information in the header of Table 1 and 2. E.g. What do you mean by „nudged runs" in this context? I guess these are the variables that you use to machine learn nudging tendencies to correct the coarse EAMxx. What is each column showing? This will strongly help readers to understand what you do and what the purpose of these two tables is. Also I would suggest to illustrate the column headers in bold font.

Line 90: Please homogenize how your are referring to the „prior FV3GFs work" throughout the manuscript. What do you mean with „coarsened reference solution evolution"? Try to clarify what you mean here, please.

Line 95: This is a little bit redundant information about the nudging time step of 3 hrs. You may very well delete this sentence.

Line 97: Please say which „model" you are meaning here, your ML model a coarse version of EAMxx?

Line 106: You need to add a link with a dedicated reference to the fv3net that follows the standards of Copernicus journals here. Please add references also throughout your manuscript when you are referring to that package.

Line 110 - 121: If these models are designed in prior published FV3GFS paper, please give references for each of the models. Something along the line „This model configuration is based on XYZ."

Line 120:  I would suggest to write „physical consistency" instead of „physical realizability" as this wording is more common in climate modeling.

Line 134: Please explain what you mean with „mass clipper" in a few words otherwise it is hard to understand for readers that are not working on convection parameterizations, what you did here and why.

Line 159: I would suggest to add a „to" between „possible" and „avoid" here.

Line 166: It would be great to show this underestimation of nudging tendency also visually in a potential supporting information figure. Are these biases occurring along the entire column or are they more pronounced on lower model levels? Also does these biases reflect some regional patterns? Such additional figures would help to improve this rather short paragraph.

Line 167-172: I would suggest to elaborate a little bit, what you are showing in Figure 2 and Figure 3. Please clearly state what is show in these figures. Is it just the difference of annual means, that looks so different or is it the mean over all differences along the time dimension. Also you need to explain how you compute the bias (please use this term also in the respective headers of the subplots) and the RMSE. Are these quantities area-weighted, taking into account Earth sphericity, or just an unweighted average over all grid cells. This needs to be clearly stated before interpreting the results. In general I would suggest to highlight that your EAMxx simulation with ML nudging introduce some regional biases. Especially the general weakening of the Intertropical convergence zone and also the reduction of precipitation along the boreal and austral storm tracks deserves a clear mention in the text.

Figure 2: You need to explain what you mean by „annual mean simulation error" as it is not introduced in the main text. Please explain all shown variables in the caption of Figure 2, e.g downwelling LW flux at the surface (first row). I would suggest to revise the second sentence as it may feel subjective given the regional biases.

Figure 3: Please revise the caption of this figures as discussed above. I would suggest to begin what is displayed here. You need to homogenize how you name the displayed difference metric, here you are writing „annual mean spatial patterns", which may confuse readers. I would strongly suggest to use one name for the water vapor path throughout the manuscript (see Line 172) and especially in this Figure. Also I would suggest to leave out any model interpretation out of the caption of a Figure. You may very well discuss these things in the main text.

Line 174-180: I like that you did these tests. Though, I would shortened this paragraph and move the largest portion of it into the supporting information or an appendix, as it is difficult to understand at this location. You may need to add more explanation of Figure 4, as it is very hard to understand, what this figure shows and also why you want to display it in your manuscript. Maybe I am missing something, but why are two random seeds using the novelty technique and one not? Why do you pick e.g. the downwelling LW flux at the surface? What is the effect of the novelty technique on other variables? If you want to keep this paragraph you need to expand the interpretation. E.g. Which dependence on random seed and the use of the novelty method is visible? Where do you see regional differences between the random seeds (I see relatively large similarities over most areas)?

Figure 4: Please revise the figure caption. Explain clearly what is shown. Here again you use a different terminology as in Figure 2 and Figure 3. Also I would personally move this figure to the supporting information as it does not show new general results or helps with the general understanding. Maybe you can move the shortened random seed paragraph and the reference to this figure to the discussion section as this would showcase that you did some sensitivity experiments? This may helps with the flow and improves the general structure of the paper.

Line 182 - 187: This paragraph is very clear and I fully agree with what you are writing. Here you may very well add that you explored the transferability of ML parameterizations from one GCM to the next, which is an important and key step for future advances in Earth System Modeling as working groups will depend on integrating ML machinery from other groups. I am thinking here of coupled simulations where people add ML atmospheric and ML ocean parameterization together. So this project was a success in my eyes as you were able to conduct stable model runs for some random seeds despite your stated model deficiencies.

Line 189-190: The point that EAMxx has no convection parameterizations is a very essential and integral point of your paper. Please try to convey this also at the begin of your results section. This will also help you to address my general circulation point above. I would suggest to add also here in the discussion regions where a usual deep convection scheme is important and hence you would expect biases of your ML algorithm.

Line 193-194: This sentence is a little bit unclear to me and needs to be revised. I guess you are stressing in the second part of the sentence, that EAMxx run on coarse resolutions need some sort of auxiliary convective parameterizations as such processes can not be directly simulated.

Line 196-199: Here you may add again a reference to one Ai2 paper, as people may be lost that are only scanning over the discussion and conclusion.

Line 199: You may delete „anecdotal"

Line 203: Which models are you meaning here, the GCM or the ML algorithm? I would call them by name, as this will help with the understanding of this paragraph.

Line 204 - 208: Please add a reference to your EAMxx github page here. It cannot hurt to remind readers that is openly accessible.

Line 212-215: Thanks for conducting these experiments. If you have results of these simulations available, you may include one figure in the supporting information that shows differences between the vertical and no vertical interpolation experiment. Again please pay attention to the use of „model" here.

Line 218 and thereafter: Please add references in the conclusion section for the readers that only have a look over the conclusion section.

Line 223: What do you mean by „doubly-periodic use cases"? You may add here an example to help to understand this sentence.

Line 231: I would include here that Kochkov24 show that "it has accurate weather forecast and improved simulation skill of some climate variables."

Line 235: You may need to revise this section to fulfill Copernicus journal standards. Just replacing all links here with citations that point in your references to the software and data would be sufficient, I guess. This webpage is helpful in this respect (https://www.geoscientific-model-development.net/policies/code_and_data_policy.html) or you may have a look over recently published GMD papers.

---

## Referee Comment (RC2)

General comments

The article presents work with the aim to use machine learning to nudge a coarse version of the EAMxx-SCREAM model towards the results of a high resolution run with the goal to obtain results corresponding to high-res runs at significantly lower computational costs. For this, the authors follow previous work with another Earth system model, FV3, and use the openly available algorithms that were successfully implemented for FV3. However, in the present article, the results of the nudging were less promising than in the previous work. The authors speculate as to why this is the case. They also argue that, despite the less effective nudging, there are additional benefits stemming from the model developments made, such as Python to C++ bridging added to the model which will also be beneficial for other developments.

In general, this is an interesting and highly timely work that should be published. I also feel very favorable to publishing work that has not proven entirely as successful as initially hoped. However, in the present form, the article feels incomplete and lacks motivation and discussion and thorough investigation as to why the work was not as effective as for the FV3 model, which would make the article useful for readers attempting similar endeavors. I therefore recommend major revisions as detailed in the appendix.

Specific comments

- Since the authors argue that there are benefits from their model developments other than the attempts at nudging to high resolution, these benefits should be stated more clearly in the motivation of the work.
- Also, it is not clear to me whether the changes made to the SCREAM code are openly available (important with respect to the benefits claimed above) – the section on Open data is very superficial.
- What is the throughput of the tested models (high res, coarse without and with nudging)?
- A detailed description of the models before ML is missing. Which parameterizations are active?
- How do the authors deal with the spin-up periods of the models? What are the initial conditions? How do they ensure that the coarse and high res models simulate the same climate state? How long were the runs for training?
- A detailed investigation of the produced nudging tendencies is missing. Here I would expect some suitable figures. Do the nudging tendencies make physical sense, considering the better resolution of physical processes in the high res? How robust are they with the different seeds or with slightly different starting conditions? How large are the nudging tendencies in comparison with the coarse unnudged tendencies? What is the consequence of nudging with 3-hour average fields?
- What motivates the use of the variables used for validation of the ML model (Tab. 2)? I notice that no dynamics variables are present despite the nudging also being applied to the winds?
- There is no detailed description of the ML models trained, please add. Please also add a description of the out-of-sample novelty detector in the methods section.
- How often is the mass clipper applied (in percent)?
- Can the authors make some sense about why they see improvements in some variables and not in others? Please add discussion here, e.g. connected to the extended discussion of the resulting nudging tendencies.
- Please add a table of all the trainings done, how many runs of them failed? (p. 8)

- Figs. 2, 3, 4 – please describe clearly what is seen in the caption, not only discussion, and make sure that the labels in the figures are correct (e.g. Delta pressure for bias in pressure)

Technical corrections

- The second sentence in the introduction (p. 1) is not a complete sentence.
- P. 1 "This project aims to develop a computationally efficient machine learning based emulator for SCREAM" seems misleading, since it is not a complete emulator of the model that is developed, just the ML-based nudging.
- P. 7 first sentence in last paragraph missing "to": While it is possible avoid … "
-
- P. 2: "both approaches have shown successful results, often outperforming state-of-the-art GCMs while only using a fraction of the computational resources" – this statement seems too simplified, please elaborate
- P. 2 First abstract in Section 2: Please add a reference for "maintains good performance across a number of high performance computing systems"